# Molecular Cloning of Toll-like Receptor 2 and 4 (*SpTLR2*, *4*) and Expression of TLR-Related Genes from *Schizothorax prenanti* after Poly (I:C) Stimulation

**DOI:** 10.3390/genes14071388

**Published:** 2023-07-01

**Authors:** Jianlu Zhang, Jiqin Huang, Haitao Zhao

**Affiliations:** 1Shaanxi Key Laboratory of Qinling Ecological Security, Shaanxi Institute of Zoology, Xi’an 710032, China; zhangjianlu@xab.ac.cn (J.Z.); huangjq1985@163.com (J.H.); 2College of Urban and Environmental Sciences, Northwest University, Xi’an 710127, China

**Keywords:** cyprinid, immune response, toll-like receptor, *Schizothorax prenanti*, polyinosinic-polycytidylic acid

## Abstract

Toll-like receptor (TLR) signaling is conserved between fish and mammals, except for TLR4, which is absent in most fish. In the present study, we aimed to evaluate whether TLR4 is expressed in *Schizothorax prenanti* (*SpTLR4*). The *SpTLR2* and *SpTLR4* were cloned and identified, and their tissue distribution was examined. The cDNA encoding *SpTLR4* and *SpTLR2* complete coding sequences (CDS) were identified and cloned. Additionally, we examined the expression levels of seven *SpTLRs* (*SpTLR2*, *3*, *4*, *18*, *22-1*, *22-2*, and *22-3*), as well as *SpMyD88* and *SpIRF3* in the liver, head kidney, hindgut, and spleen of *S. prenanti*, after intraperitoneal injection of polyinosinic-polycytidylic acid (poly (I:C)). The *Sp*TLR2 and *Sp*TLR4 shared amino acid sequence identity of 42.15–96.21% and 36.21–93.58%, respectively, with sequences from other vertebrates. *SpTLR2* and *SpTLR4* were expressed in all *S. prenanti* tissues examined, particularly in immune-related tissues. Poly (I:C) significantly upregulated most of the genes evaluated in the four immune organs compared with the PBS-control (*p* < 0.05); expression of these different genes was tissue-specific. Our findings demonstrate that TLR2 and TLR4 are expressed in *S. prenanti* and that poly (I:C) affects the expression of nine TLR-related genes, which are potentially involved in *S. prenanti* antiviral immunity or mediating pathological processes with differential kinetics. This will contribute to a better understanding of the roles of these TLR-related genes in antiviral immunity.

## 1. Introduction

The immune system of vertebrates includes the innate and adaptive immune systems, which are essential in fish immunity [1]. In fish, innate pattern recognition receptors (PRRs) activate the innate immune response through a series of highly conserved pathogen-associated molecular patterns [2]. The PRRs in fish include the RIG-I-like receptor, NOD-like receptor, C-type lectin receptor, and the toll-like receptor (TLR) family [3]. Among the PRR families, the TLR family is the most widely studied [4]. These receptors were first identified in *Drosophila melanogaster* in 1985 [5]. The TLR family is divided into six subfamilies, based on evolutionary relatedness: TLR1, 3, 4, 5, 7, and 11 subfamily. To date, at least 22 TLRs have been cloned and identified in bony fish (TLR1, 2, 3, 4, 5M, 5S, 7, 8, 9, 13, 14, and 18-28), some of which are bony fish-specific TLRs, such as TLR18-28 [6,7]. In this study, another two TLRs, the *Sp*TLR2 (belonging to the TLR1 subfamily) and *Sp*TLR4 (belonging to the TLR4 subfamily) were cloned and identified. Gene cloning and functional identification of fish-specific TLRs have also become research hotspots. Studying fish TLRs is necessary for understanding the immune system of lower vertebrates.

TLRs play crucial roles in the identification of microbial pathogens that infect fish. Together with myeloid differentiation factor 88 (MyD88), interferon regulatory factors (IRFs), and other factors in the immune signaling pathway, TLRs are involved in the identification of most pathogenic microorganisms, including bacteria, viruses, and parasites [8,9,10,11]. The adapter molecules are recruited by the toll/IL-1 receptor (TIR) domain of TLR during TLR signal transduction, leading to the activation of diverse signaling pathways. These signaling pathways involving TLRs can be divided into two categories: MyD88-dependent and MyD88-independent pathways. In the former pathway, MyD88 acts as an adaptor protein and is recruited by TLRs as the first signaling protein, playing a key role in TLR signal transduction [12,13,14]. The MyD88-independent pathway is a specific signaling pathway involving only a few TLRs, mainly related to antiviral signaling, also known as the TIR-domain-containing adaptor-inducing interferon (IFN)-β-dependent pathway [7,10]. For example, TLR3 and TLR22 activate IRF3 and IRF7 to complete the immune response via a TIR-domain-containing adaptor-inducing IFN-β-dependent pathway [2,15]. IRF family members (IRF1-11) have immunoregulatory functions; IRF3 plays a crucial role in the innate immune resistance system against viruses [16] and in the MyD88-independent pathway [17].

The immune-related components of teleost fish differ from those in mammals [18]. TLR4, an ancient TLR and the only member of the TLR4 subfamily [6], is present in mammals but not in all fish. The first mammalian TLR4 discovered was human TLR4, which is thought to be homologous to TLR1 in *D. melanogaster* [19]. In fish, TLR4 is mostly found in cyprinids, having been identified in zebrafish (*Danio rerio*) [20], rohu (*Labeo rohita*) [21], rare minnow (*Gobiocypris rarus*) [22], common carp (*Cyprinus carpio*) [23], grass carp (*Ctenopharyngodon idella*) [24], and Przewalski’s naked carp (*Gymnocypris przewalskii)* [25]. Additionally, TLR4 is also expressed in other fish families such as channel catfish (*Ictalurus punctatus*) [26] and blunt snout bream (*Megalobrama amblycephala*) [27]. Conversely, TLR4 is not found in most fish [6,28] and is absent in spotted green pufferfish (*Tetraodon nigroviridis*) [29] and pufferfish (*Fugu rubripes*) [30]. In the gilthead seabream (*Sparus aurata*), the presence of a TLR4 ortholog is unknown [29].

Prenant’s schizothoracin (*Schizothorax prenanti*) belongs to the fish family Cyprinidae, known locally as “yang-fish” in Hanzhong city (Shaanxi, China) or “ya-fish,” together with *S. davidi*, in Ya’an city (Sichuan, China). As rare and high-quality cold-water fish in production areas, economically important fish have been artificially cultivated and marketed for consumption at approximately 120 yuan/kg [31]. Because of intensive feeding, yang-fish are susceptible to bacterial infections, such as *Aeromonas hydrophila* [32,33] or *Streptococcus agalactiae* [34], as well as reoviruses [35], which hinder the healthy development of yang-fish farming.

In this study, we aimed to evaluate the expression of TLR2 and TLR4 in *S. prenanti* in physiological conditions and after induction of antiviral response mechanisms with a poly (I:C) challenge. Poly (I:C) is a viral analog that has been shown to trigger innate and adaptive immune responses involving TLR signaling, depending on the species [36,37,38]. We first confirmed the existence of *TLR4* in yang-fish and its secondary structure composition; furthermore, we predicted the 3D-structural models of *Sp*TLR4 and *Sp*TLR2 proteins. The expression patterns of TLR-related genes in different immune organs (liver, head kidney, hindgut, and spleen), in response to poly (I:C), were analyzed by quantitative real-time PCR (qRT-PCR). The genes we analyzed included *SpTLR2*, *3*, *4*, *18*, *22-1*, *22-2*, *22-3*, *SpIRF3*, and *SpMyD88*. Our findings contribute to further clarifying the roles of *SpTLR*s, *SpIRF*3, and *SpMyD88* in the immune mechanisms of fish and to a better understanding of the function of *Sp*TLR4.

## 2. Materials and Methods

### 2.1. Animal Treatments

Healthy *S. prenanti* (121.7 ± 28 g) were purchased from the Qunfu Yang-fish professional breeding cooperative (Hanzhong, China). The experimental fish were kept in glass tanks with a volume of (60 × 30 × 40) cm^3^ with aerated tap water at a temperature of 20 ± 1 °C. Twelve fish were placed per tank, and feed conditions for the fish refer to our previous research [31] After 10 days of acclimatization, *S. prenanti* in the test group were stimulated with intraperitoneal injection of poly (I:C) (P1530, Sigma-Aldrich, St. Louis, MO, USA) at a dose of 5 mg/kg body weight. Fish in the control group were injected with the same dose of phosphate-buffered saline (PBS). To evaluate the expression of *SpTLR2*, *3*, *4*, *18*, *SpTLR22*s (*22-1*, *22-2*, *22-3*), *SpIRF3*, and *SpMyD88* under poly (I:C) stimulation, anatomical samples of the poly (I:C)-injected animals were obtained at 12 and 24 h after infection (4 animals per time point). Four PBS-injected fish were used as controls and their tissues were collected at the 24 h. The fish were anesthetized with 80 mg/L eugenol (Daoyuan Biotechnology Co. LTD, Guangzhou, China) for 3 min before dissection. The heart, head kidney, liver, hindgut, intraperitoneal fat, muscle, and spleen were sampled and preserved in liquid nitrogen.

### 2.2. RNA Extraction and cDNA Synthesis

Tissue total RNA was extracted using TRIzol reagent (Invitrogen, Waltham, MA, USA). The concentration of total RNA and purity was determined with agarose gel electrophoresis, and an A260/280 ration was determined using a Nanodrop One spectrophotometer (Thermo Fisher Scientific, Waltham, MA, USA). The cDNA was synthesized from total RNA with the RevertAid First Strand cDNA Synthesis Kit (Thermo Fisher Scientific) following to the manufacturer’s instructions.

### 2.3. CDS Cloning of SpTLR2 and SpTLR4

Based on the transcriptome sequencing of *S. prenanti* and the sequences from Cyprinidae fish, specific TLR2 and TLR4 primers were designed (Table 1) with the help of PrimerQuest. The primers were synthesized by Tsingke Biotechnology Co., Ltd. (Xi’an, China). The *SpTLR2* and *SpTLR4* genes were amplified using PrimerStar Max DNA polymerase (Takara Bio, Shiga, Japan) with spleen cDNA as template. The PCR was carried out as follows: 35 cycles of denaturing under 98 °C for 10 s, annealing under 50 °C for 15 s, and extending under 72 °C for 40 s. A tailing was added to the 3’ end of the PCR product with the DNA A-Tailing Kit (Takara Bio). The products were ligated with pMD19-T vector (Takara Bio) and then transformed into competent cells of *Escherichia coli* DH5α (TIANGEN, Beijing, China). Then competent cells were cultured on an LB agar plate (containing 100 mg/L ampicillin) at 37 °C. Subsequently, positive bacterial clones were sequenced to confirm the cloning.

### 2.4. Sequence Analysis

The *SpTLR2* and *SpTLR4* complete CDS were identified with ORF Finder (http://www.ncbi.nlm.nih.gov/orffinder/, accessed on 16 February 2023). Several *Sp*TLR2 and *Sp*TLR4 properties were predicted using diverse online tools: the molecular weight and isoelectric point (https://web.expasy.org/compute_pi/, accessed on 16 February 2023); signal peptides, leucine-rich repeats (LRRs), transmembrane domains, and TIR domains (http://smart.emblheidelberg.de/, accessed on 25 February 2023); secondary structures (http://bioinf.cs.ucl.ac.uk/psipred, accessed on 22 February 2023); and 3D structures (https://swissmodel.expasy.org/, accessed on 16 February 2023). Multiple amino acid sequences were aligned with Clustal X2 [39], and the phylogenetic trees of TLR2 and TLR4 from different vertebrates were constructed using the neighbor-joining method with the MEGA 11.0 software [40].

### 2.5. Tissue Distribution of SpTLR2 and SpTLR4 mRNA

Total RNA was isolated from the different tissues (heart, liver, spleen, intraperitoneal fat, head kidney, muscle, and hindgut) and cDNA was prepared as described. A qRT-PCR was performed using FastStart Essential DNA Green Master (Roche, Basel, Switzerland) on Applied Biosystems Step One Plus (Life Technologies, Carlsbad, CA, USA). The primers used in this study are listed in Table 1. *S. prenanti*-specific actin primers were used as an internal control. Triplicat analyses of *SpTLR2*, *SpTLR4*, and actin mRNA expression were performed for all samples, and the data were analyzed according to the 2^−ΔΔCT^ method [41]. Tissues collected from poly (I:C)-challenged animals were processed and analyzed using the same methods to assess the changes in the expression levels of TLR-related genes, induced by the viral analog.

### 2.6. Statistical Analysis

SPSS 22.0 (IBM Corp., Armonk, NY, USA), and GraphPad Prism 5 (GraphPad Software Inc., San Diego, CA, USA) software were performed for data analysis and visualization, respectively. The mRNA expression abundance was analyzed using a one-way analysis of variance. All data are presented as the mean ± standard error (*n* = 4); statistical significance was established as *p* < 0.05.

## 3. Results

### 3.1. Identification and Structural and Phylogenetic Analysis of SpTLR2 and SpTLR4

First, we set out to confirm the expression of *TLR2* and *TLR4* in *S. prenanti*. We observed the expression of both genes and investigated their features. We found that the CDS length of *SpTLR2* was 2379 bp (GenBank accession no. OQ676992), and the predicted *SpTLR2* ORF encoded a protein with 792 amino acids. The calculated molecular mass and theoretical isoelectric point of *Sp*TLR2 was 199.19 kDa and 4.89, respectively. Domain architecture analysis of *Sp*TLR2, using the SMART tool (http://smart.emblheidelberg.de/, accessed on 28 February 2023), revealed the presence of canonical structural motifs in TLR family proteins. These include a signal peptide, six LRRs, and one TIR domain; this *Sp*TLR2 domain structure is similar to that of other vertebrate TLR2s (Figure 1).

For *SpTLR4*, we found that the CDS length was 2343 bp (GenBank accession no. OQ108869). The predicted *SpTLR4* ORF encoded a 780 amino acid protein, and its calculated molecular mass and theoretical isoelectric point were 197.68 kDa and 4.93, respectively. Using the SMART tool, we identified the following structural domains in *Sp*TLR4: six LRRs, one transmembrane domain, and one TIR domain (Figure 2). The TLR4 domain regions in other vertebrates are shown in Figure 2. Figure 3 shows the secondary structure and predicted 3D structure of *Sp*TLR2 (Figure 3a,b, respectively) and *Sp*TLR4 (Figure 3c,d, respectively). Similar to S*p*TLR2, *Sp*TLR4 has a horseshoe-shaped solenoid structure with parallel β-sheet lining the inner circumference and α-helices flanking its outer circumference.

To infer the evolutionary relationships between *Sp*TLR2 and *Sp*TLR4, a phylogenetic tree was constructed based on the alignment of *Sp*TLR2 and *Sp*TLR4 amino acid sequences with other available vertebrate amino acid sequences for these two proteins. The *Sp*TLR2 amino acid sequence was most similar to that of fish and was closest to the golden mahseer (*Tor putitora*) TLR2, with 96.21% identity. We analyzed the phylogeny of the *Sp*TLR2 and *Sp*TLR4 amino acid sequences to determine the relationships between *S. prenanti* and other vertebrates based on sequences in the GenBank database (Figure 4). The results revealed a high TLR2 and TLR4 amino acid sequence identity between *S. prenanti* and the fish of the cyprinid family, to which both *S. prenanti* and *T. putitora* belong to. Similar results were obtained for *Sp*TLR4.

### 3.2. Tissue Distribution of SpTLR2 and SpTLR4 Expression

We quantified *SpTLR2* and *SpTLR4* mRNA expression in the eight tissues (heart, head kidney, spleen, liver, muscle, gill, hindgut, and intraperitoneal fat) of 4 fish using qRT-PCR to determine the transcript abundance of both TLRs. The liver, head kidney, spleen, and hindgut of fish are generally regarded as immune organs that mediate the immune response [42]. The mRNA abundance of β-actin was used for normalization. The expression of splenic *SpTLR2* was the highest, followed by the heart and intraperitoneal fat. Conversely, *SpTLR2* levels in the head kidney, hindgut, muscle, and liver were significantly lower (*p* < 0.05), except in the gills where no significant differences were found. In contrast, *SpTLR4* was found to be most abundant in the liver, in which its expression was significantly higher than that for the other seven tissues (*p* < 0.05); the spleen had the second-highest tissue expression of *SpTLR4*, which was much higher than those in the other six tissues (*p* < 0.05). Moreover, the *SpTLR4* level in the heart was more pronounced and higher than those in the intraperitoneal fat, head kidney, and muscle tissues (*p* < 0.05), (Figure 5).

### 3.3. Expression of TLR-Related Genes Following Poly (I:C) Challenge

To determine the changes in *TLR2*, *3*, *4*, *18*, *TLR22s* (*22-1*, *22-2*, and *22-3*), *MyD88*, and *IRF3* in *S. prenanti* tissues at 12 and 24 h after poly (I:C) stimulation, the mRNA levels of the genes were quantified using qRT-PCR in the liver, head kidney, spleen, and hindgut tissues. The results are shown in Figure 6.

#### 3.3.1. Expression of *SpTLR2*

The *SpTLR2* transcripts in the head kidney increased significantly at the 12 h time point (*p* < 0.001). Conversely, at the 24 h time point, the level of *SpTLR2* was significantly decreased relative to both the control and to the poly (I:C) 12 h conditions (*p* < 0.05 and *p* < 0.001, respectively). Similarly, the expression level of hepatic *SpTLR2* at 24 h was significantly lower than that in the control group (*p* < 0.05). In the hindgut, the *SpTLR2* mRNA was pronounced and upregulated at 24 h after poly (I:C) stimulation (*p* < 0.001). Conversely, the spleen was the only organ in which the levels of *SpTLR2* were downregulated at both 12 and 24 h post-poly (I:C) challenge (*p* < 0.001), (Figure 6a).

#### 3.3.2. Expression of *SpTLR3*

In the liver, *SpTLR3* levels were significantly higher at both time points than those in the PBS-injection control (*p* < 0.01 at 12 h; *p* < 0.001 at 24 h), but no significant difference was found between the two poly (I:C) challenge groups. Similar to the effects observed for *SpTLR2* in the hindgut, the expression level of the *SpTLR3* gene in this organ was significantly upregulated at 12 h, relative to the control (*p* < 0.01), and downregulated at 24 h, relative to both the PBS-injected group (*p* < 0.001) and the 12 h poly (I:C)-injection group (*p* < 0.001). In contrast, *SpTLR3* mRNA levels were not significantly different in the head kidney or spleen following poly (I:C) injection (Figure 6b).

#### 3.3.3. Expression of *SpTLR4*

The relative transcript level of *SpTLR4* was generally higher than that of most other genes examined. The highest expression level of *SpTLR4* was detected in the spleen, at 24 h after poly (I:C) injection (approximately 113-fold of the transcript level in the PBS-injected group; *p* < 0.001). In addition, the level of hepatic *SpTLR4* was significantly upregulated at 12 (*p* < 0.01) and 24 h (*p* < 0.001) post-poly (I:C) stimulation. In both the head kidney and hindgut, *SpTLR4* mRNA was significantly upregulated at 12 h (*p* < 0.01) and 24 h (*p* < 0.05), compared to the PBS-control; when compared to the 12 h time point, the level of *SpTLR4* at 24 h was slightly decreased (*p* < 0.05) in the head kidney (Figure 6c).

#### 3.3.4. Expression of *SpTLR22s*

Compared with the PBS control, the relative expression of *SpTLR22-1* mRNA at 12 h post-poly (I:C) injection was unchanged in the liver, head kidney, and spleen; at this time point, the only organ showing a significant increase in the level of this transcript was the hindgut (*p* < 0.05). In contrast, at the 24 h time point, all four organs displayed significant changes in the levels of *SpTLR22-1* mRNA (upregulated: liver (*p* < 0.05), hindgut (*p* < 0.001), and spleen (*p* < 0.05); downregulated: head kidney (*p* < 0.01)) (Figure 6d). In the head kidney and hindgut, the temporal pattern of expression was similar to that of the *SpTLR22-1* transcript (downregulated at 24 h post-injection and upregulated at 12 and 24 h, respectively). Conversely, in the spleen, the transcript was significantly downregulated at both time points (*p* < 0.001), whereas the level of hepatic *SpTLR22-2* was only downregulated at 24 h (*p* < 0.05) after the poly (I:C) injection (Figure 6e). The expression levels of *SpTLR*22-*3* followed a different pattern than that of *SpTLR*22-*1* or *SpTLR*22-*2*. The expression level of *SpTLR22-3* was unchanged at either time point in the liver and hindgut. In the head kidney, however, *SpTLR22-3* mRNA was significantly increased at both 12 (*p* < 0.01) and 24 h (*p* < 0.001) after the poly (I:C) injection. Moreover, the level of splenic *SpTLR*22-3 was significantly increased but only at the 24 h point (*p* < 0.001) (Figure 6f).

#### 3.3.5. Expression of *SpTLR18*

The levels of hepatic *SpTLR18* were most significantly upregulated at both 12 and 24 h relative to the PBS-injection group (*p* < 0.001). Additionally, the expression at 24 h was also significantly higher than that at 12 h (*p* < 0.01). At 12 h post-poly (I:C) challenge, *SpTLR18* expression remained unchanged in the kidney, hindgut, and spleen. However, at 24 h, the levels of *SpTLR18* were significantly downregulated in the head kidney (compared to PBS, *p* < 0.05; and to the 12 h, *p* < 0.01), upregulated in the hindgut (*p* < 0.01), and remained unchanged in the spleen (Figure 6g).

#### 3.3.6. Expression of *SpMyD88*

The relative levels of *SpMyD88* in the liver and hindgut were unchanged. Only the head kidney and spleen displayed different temporal expression patterns of this transcript. Compared to the PBS group, the levels of *SpMyD88* in the head kidney were extremely upregulated at 24 h after poly (I:C) stimulation (*p* < 0.05). The levels of *SpMyD88* mRNA in the spleen were significantly upregulated at the 12 h time point (*p* < 0.05) and then significantly downregulated at 24 h compared to the 12 h stimulation group (*p* < 0.001) and the PBS-injection group (*p* < 0.01) (Figure 6h).

#### 3.3.7. Expression of *SpIRF3*

The *SpIRF3* transcripts were upregulated in all four tissues at both time points after the poly (I:C) treatment. The expression of hepatic *SpIRF3* at 12 and 24 h was pronounced higher than that in the PBS control (*p* < 0.05 and *p* < 0.01, respectively). In the head kidney, *SpIRF3* mRNA was significantly upregulated at both time points but lower in the 24 h condition, relative to the 12 h post-poly (I:C) group (*p* < 0.001). The temporal expression patterns of *SpIRF3* in the hindgut and spleen were similar; both tissues displayed increased levels in this transcript at 12 h, which remained stable at 24 h the after poly (I:C) injection (hindgut, *p* < 0.01; spleen, *p* < 0.001). Additional time points would be required to establish whether these transcripts’ expression reached a peak in these organs (Figure 6i).

## 4. Discussion

Our study firstly identified the presence of TLR2 and TLR4 in *S. prenanti*. The predicted *Sp*TLR2 and *Sp*TLR4 amino acid sequence we describe in this study includes the typical conserved structure of the TLR protein family. Previous studies have confirmed that the LRR domains in TLR proteins are related to the identification of pathogen components and that the number of LRR domains present in the protein sequence varies in different animals [43,44]. *Sp*TLR3, *Sp*TLR5, *Sp*TLR22, and *Sp*TLR25 contain a signal peptide [18,45,46,47], as does *Sp*TLR2. However, we found no signal peptide in *Sp*TLR4 in this study, which is consistent with TLR4 proteins from other species: TLR4.1, TLR4.2 from *Ctenopharyngodon idella* [24], and TLR4a from *D. rerio* [29]. The absence of the signal peptide suggests that these proteins may play a role in the cytoplasm. TLR4 is not expressed in most fish species, probably because of the diversity of environments in which they live and their evolutionary history [28]. Contrary to the mammalian protein, *D. rerio* TLR4 cannot recognize lipopolysaccharide (LPS) [29], thus indicating that some species of fish have different subtypes of the TLR4 protein, some of which may only function as cytoplasmic pattern recognition receptors.

The main function of IRFs is to interfere with viral replication by inducing the production of IFN [48,49]. Some IRFs, such as IRF3 and IRF9, activate IFN-α/β and their downstream pathways in the host’s antiviral immune process [50,51]. In fish, *IRF3* was first detected in rainbow trout (*Oncorhynchus mykiss*), and its expression was found to be induced after treatment with poly (I:C) [52]. In Atlantic salmon (*Salmo salar*), MyD88 interacts with IRF3 and IRF7 to regulate the IRF-induced IFN response [53]. Moreover, *IRF3* overexpression greatly induces the transcriptional activity of *IFN*, and the transcription of type I *IFN* was regulated by *IRF3* after challenged by a double-stranded virus [54]. In this study, we found that *SpIRF3* was upregulated in all four tissues examined, especially in the head kidney and liver. These results lend support to the antiviral role exerted by fish IRF3.

MyD88 plays a key role in the transduction of TLR-mediated signaling and is frequently evaluated in studies investigating signaling pathways involving TLRs. After yellow drum (*Nibea albiflora*) [55] and Japanese flounder (*Paralichthys olivaceus*) were treated with *Pseudomonas plecoglossicida* and *Edwardsiella tarda*, respectively, *NaMyD88* and *PcMyD88* were extremely raised in the kidney and spleen, compared to the expression levels in the corresponding control groups [56]. To date, few studies have evaluated the effect of viruses on *MyD88* expression in fish. Of the few available reports, most have focused on changes in the expression level of *MyD88* after stimulation with the viral nucleic acid analog poly (I:C). *MyD88* levels in the blood cells of *Litopenaeus vannamei* are lower than those in control conditions, except at 4 h and 12 h after poly (I:C) stimulation. Conversely, white spot virus (dsDNA virus) significantly increases the expression level of *LvMyD88* [57], and SAV3 (ssRNA virus) upregulates the level of *MyD88* in *S. salar* spleen, which remains elevated for 28 days [58]. Similar results are presented in this study, whereas the timing of the immune response varies depending on the pathogen and the species of fish.

To date, 22 TLRs have been identified in bony fish belonging to six TLR subfamilies: TLR1 (TLR1, 2, 14, 18 (fish-specific), 24, 25, 27, and 28), TLR3 (TLR3), TLR4 (TLR4), TLR5 (TLR5M and 5S), TLR7 (TLR7, 8, and 9), and TLR11 (TLR13, 19, 20, 21, 22, 23, and 26) [7,28,59]. TLR2 forms homodimers or heterodimers with TLR1 and TLR6, recognizes various ligands from bacteria, and participates in viral recognition. TLR2 from *Epinephelus coioides* participates in the immune response to anti-LPS and poly (I:C) [60]. In the early stage of viral hemorrhagic septicemia virus (VHSV) infection in olive flounder (*P. olivaceus*), *TLR2* and *IRF3* are significantly upregulated. Accordingly, we observed that *SpTLR2* and *SpTLR*18 are upregulated following a poly (I:C) challenge, particularly in the head kidney at 12 h and hindgut at 24 h. Our previous study demonstrated that LPS significantly increases *SpTLR18* levels [46], and the results of the present study support the likely role of this protein in the innate immune responses of bony fish. Moreover, in the spleen, the expression level of *SpTLR2* was significantly downregulated compared to that in the PBS-control group at 12 and 24 h, and the *SpTLR18* level was unchanged at these two time points. Based on these observations, we speculate that the upregulation of *SpTLR2* and *SpTLR18* after poly (I:C) induction may occur at intermediate or later time points; however, validation of this hypothesis requires further investigation.

TLR3 is the single member of the TLR3 subfamily. In mammals, TLR3 mediates the antiviral immune response to dsRNA viruses, which is similar to its function in fish. Studies have shown that fish *TLR3* genes were significantly upregulated in immune-related tissues and organs infected with viruses or poly (I:C), including *D. rerio* infected with VHSV [61], renal leukocytes from rainbow trout [62] and large yellow croak (*Pseudosciaena crocea*) [63], and *G. rarus* infected with grass carp reovirus (GCRV) [64]. In this study, *SpTLR3* transcripts in the liver and hindgut were significantly upregulated 12 h after poly (I:C) induction. These results suggest that fish TLR3 recognizes viruses and plays an important role in the immune response.

TLR22 is another fish-specific TLR belonging to the TLR11 subfamily that was first discovered in goldfish in 2003 [65]. Subsequently, TLR22 has been cloned and identified in 17 fish species, including *D. rerio* [20], *P. olivaceus* [66], *S. salar* [67], *Fugu rubripes* [68], *Pseudosciaena crocea* [69], *C. idella* [70], *Epinephelus coioides* [71], *Gadus morhua* [72], *I. punctatus* [26], *L. rohita* [73], *C. mrigala* [74], *S. aurata* [75], *Scophthalmus maximus* [76], *Seriola lalandi* [77], *C. carpio L.* [78], and *S. prenanti* [18]. Initially, two subtypes of TLR22 (named *TLR22-1* and *TLR22-2*) were discovered in rainbow trout, which have highly similar functions and were called ‘twin’ TLRs. Subsequently, *TLR22-1*, *-2*, and *-3* were identified in *S. salar* (GenBank accession no.: AM233509, FM206383, and BT045774, respectively). These reports, as well as the results from the present study, suggest that fish TLR22 is a multifunctional immune receptor involved in the defense and immune response of almost all pathogenic microorganisms; however, the corresponding recognition mechanisms and downstream signaling pathways remain unclear. To date, only two studies have explored the downstream signaling pathways mediated by TLR22. In the first report, TLR22 of *T. rubripes* was demonstrated to be located in the cell membrane and to induce IFN expression in response to viral infection [68]. In contrast, another report showed *Ec*TLR22 to be located in the endosome and to mediate protective mechanisms, inhibiting the transmission of antiviral and inflammatory signals to prevent excessive inflammation [79]. These results suggest that TLR22 may have different functions in different fish species. Therefore, additional studies are needed to shed light on the signaling mechanism mediated by TLR22. A previous study reported that the mRNA levels of *SpTLR22-1* in the head kidney and spleen were upregulated 12 h after a poly (I:C) challenge, while *SpTLR22-3* significantly increased at both 12 and 24 h points in the head kidney; conversely, *SpTLR22-2* did not change at either the 12 or 24 h time points [18]. The results described here show some differences relative to this previous report. We found that the transcripts of *SpTLR22-3* were much higher than that of *TLR22-1* and *-2*, especially in the head kidney and spleen, at 24 h after infection. Our findings suggest that the ‘triplet’ S*pTLR22*s (*TLR22-1*, *-2*, and *-3*) jointly mediate the recognition of poly (I:C) and are involved in the immune response.

The biggest difference in TLR-signaling pathways between mammals and fish pertains to the TLR4-mediated signaling pathway [80]. TLR4 is a direct receptor of bacterial LPS [2]. Unlike in mammals, TLR4 is absent in most fish and is mainly found in cyprinids. This discrepancy in protein expression raises the question of whether mammal and fish TLR4 have similar functions in viral recognition. In *C*. *idella* infected with GCRV, the expression of TLR4 has been reported to be increased in the muscle and liver [24]. Similar results have been observed in *G*. *rarus* infected with GCRV [22]. Our present findings support these previous reports. We found that the relative transcript level of *SpTLR4* was generally higher, compared to the other genes examined, and that its expression was highest in the spleen at 24 h after the poly (I:C) stimulation relative to the control group. These results suggest that fish TLR4 expression is induced in response to viral infection and may play a crucial role in the immune response not just in antimicrobial immunity. However, its ligand specificity and function require further study. Overall, our study found evidence of the following: TLR4 is present in *S*. *prenanti*; *SpTLR4* is involved in antiviral immunity; the spleen is the most sensitive immune organ for *SpTLR4* detection at 24 h following the poly (I:C) injection. Of the nine genes examined in this study, the upregulation of *SpTLR4* expression was higher, especially for the level in the spleen at 24 h which significantly increased 110-fold. In addition, the *SpTLR3* and *SpTLR18* in the spleen and *SpTLR22-3* and *SpMyD88* in both the liver and hindgut were noninducible by poly (I:C), and the other genes in the four immune tissues were inducible by poly (I:C) in this study.

## 5. Conclusions

In this study, the CDS of *SpTLR2* and *SpTLR4* were successfully cloned and characterized. Phylogenetic analysis showed that *Sp*TLR2 and *Sp*TLR4 proteins were most closely related to TLR2 and TLR4 from golden mahseer. Multiple sequence alignment showed that *Sp*TLR2 and *Sp*TLR4 are moderately conserved. These two proteins were expressed in all tissues examined; *SpTLR2* was the most abundantly expressed in the spleen and *SpTLR4* in the liver. The poly (I:C) challenge affected the expression of several TLR-related genes in an organ-specific manner, suggesting their involvement in antiviral immunity or pathological processes. Overall, our findings demonstrate that *SpTLR2* and *SpTLR4* are likely involved in the immune response. These findings contribute to a better understanding of the mechanisms of immunity in lower vertebrates, which may shed light on response mechanisms to infections in economically relevant fish species.

## Figures and Tables

**Figure 1 genes-14-01388-f001:**
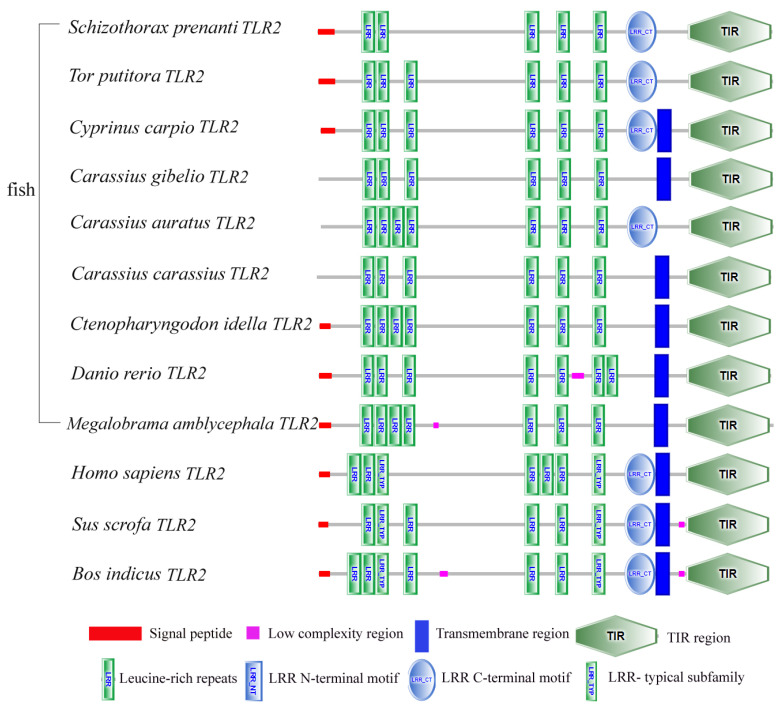
Comparison of TLR2 domain structures in *S. prenanti* and other vertebrates. [Species name (GenBank accession number)]: *Tor putitora* (UFE16643), *Cyprinus carpio* (ACP20793), *Carassius gibelio* (AGR53440), *Carassius auratus* (QHZ60136) and *Carassius carassius* (AGO57934), *Ctenopharyngodon idella* (ACT68333), *Danio rerio* (NP_997977), *Megalobrama amblycephala* (ANI19836), *Homo sapiens* (NP_001305716), *Sus scrofa* (NP_998926), and *Bos indicus* (ALZ41705).

**Figure 2 genes-14-01388-f002:**
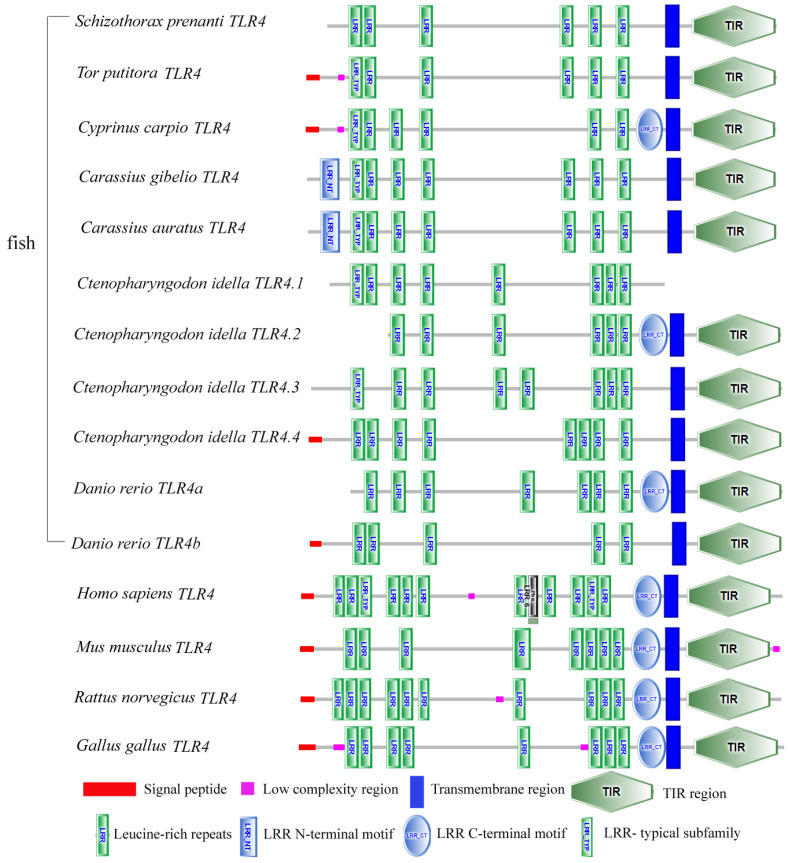
Comparison of TLR4 domain structures in *S. prenanti* and other vertebrates. Pfam: LRR_6 represents this is a Pfam domain [Species name (GenBank accession number)]: *T. putitora* (BBI01014), *C. carpio* (NP_042592483), *C. gibelio* (XP_052427945), *C. auratus* (XP_026080748), *C. idella* TLR4.1 (AEQ64877), *C. idella* TLR4.2 (AEQ64878), *C. idella* TLR4.3 (AEQ64879), *C. idella* TLR4.4 (AEQ64880), *D. rerio* TLR4a (ACE74929), *D. rerio* TLR4b (AAQ90475), *H. sapiens* (AAF05316), *Mus musculus* (EDL31078), *Rattus norvegicus* (QQJ42887), and *Gallus* (AJR32867).

**Figure 3 genes-14-01388-f003:**
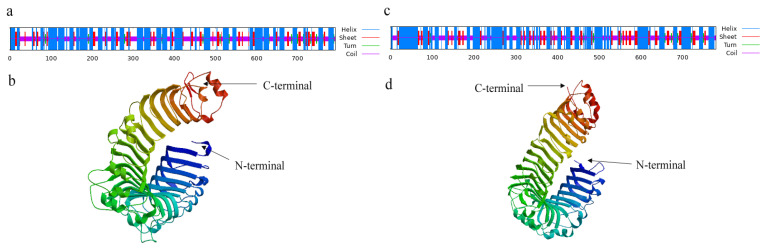
Predicted secondary structure and 3D structural models of the *Sp*TLR2 and *Sp*TLR4 proteins. (**a**) Predicted secondary structure of *Sp*TLR2 protein. (**b**) 3D structures of predicted *Sp*TLR2 protein. (**c**) Predicted secondary structure of *Sp*TLR4 protein. (**d**) 3D structures of predicted *Sp*TLR4 protein.

**Figure 4 genes-14-01388-f004:**
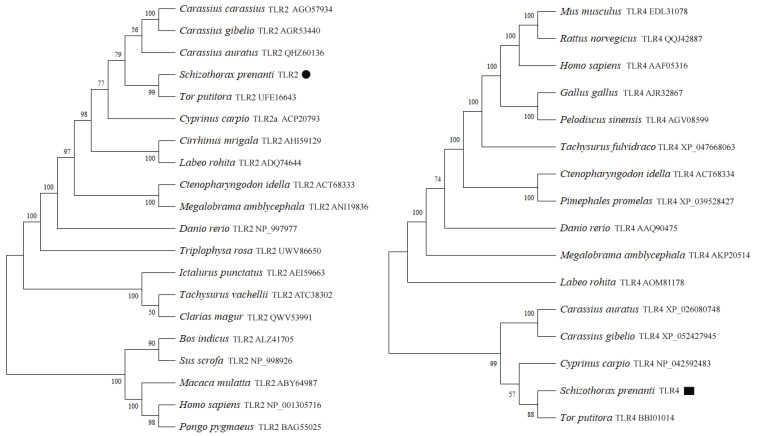
Phylogenetic tree of relationships between *Sp*TLR2/*Sp*TLR4 and other vertebrates. The tree was constructed using the neighbor-joining method with the MEGA 11.0 software. Numbers at nodes indicate proportions of bootstrapping after 1000 replications. ● *S. prenanti* TLR2, ■ *S. prenanti* TLR4.

**Figure 5 genes-14-01388-f005:**
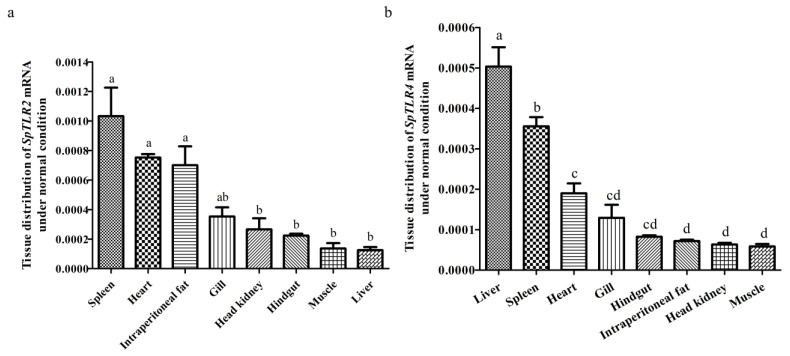
Abundance of *SpTLR2* (**a**) and *SpTLR4* (**b**) transcripts under normal conditions in *S. prenanti* spleen, heart, intraperitoneal fat, gill head kidney, hindgut, muscle, and liver tissues, as determined using qRT-PCR. The loading control was β-actin. a, b, c, and d means with different letters are significantly different from each other (*p* < 0.05).

**Figure 6 genes-14-01388-f006:**
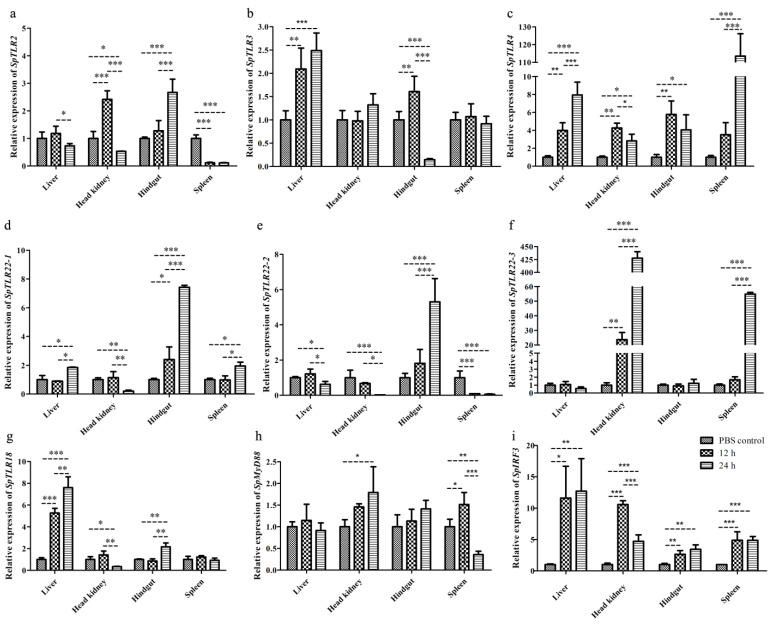
Relative levels of *SpTLR2* (**a**), *SpTLR3* (**b**), *SpTLR4* (**c**), *SpTLR22-1* (**d**), *SpTLR22-2* (**e**), and *SpTLR22-3* (**f**), *SpTLR18* (**g**), *SpMyD88* (**h**) and *SpIRF3* (i) in *S. prenanti* liver, head kidney, hindgut, and spleen at 12 and 24 h point after poly (I:C) injection. Values were normalized using β-actin. Statistically significant differences between the groups are marked with asterisks (* *p* < 0.05, ** *p* < 0.01, *** *p* < 0.001).

**Table 1 genes-14-01388-t001:** Primers for *SpTLR2* and *SpTLR4* cloning and qRT-PCR.

Primers	Sequences (5′–3′)	Annealing Temperature (°C)	Size (bp)
Primers for CDS cloning			
TLR2-F	TTAATGGCAGTCAGGATGAG	50	2447
TLR2-R	ACATTGCGTTTAGGTACTTGG		
TLR2 walking 1	CCATGCGATCGAACAGGTCT	For sequencing	
TLR2 walking 2	AATTGGTGCGCGCCTATTTC		
TLR4-F TLR4-R	ATGACCTCAAACAAGGCTGGC	50	2634
	AATGTAAAACCATACTGCCAT		
TLR4 walking	GATGCTGACGATGTTCCGGA	For sequencing	
Primers for qRT-PCR			
TLR2-F	GATCAACGGCACAGTGTTTG	62	170
TLR2-R	CAGGTCTGAAAGGAGGTTCTG		
TLR3-F	GCTGAAAGGAGATGAGTTAGAG	62	110
TLR3-R	ACGTAGGGACATGGATGAA		
TLR4-F	CTTGGTGTCGCTTTGAGTTTG	62	107
TLR4-R	GTCTCTGCTCCACTTTAGGTATG		
TLR18-F	ACAGACTAAATGGCCAGGGAAG	62	118
TLR18-R	AACCACAAGCAAGGGCAAAG		
TLR22-1-F	CCTCTTCTTAGCCTTCCTTTAC	62	94
TLR22-1-R	CTCGTCTTTGGTGTTGTAGG		
TLR22-2-F	TTCCAGGGACTGTGGTATTTG	62	98
TLR22-2-R	GCCCACAGATAAGGAGTGTAAG		
TLR22-3-F	CCATCGGCATTCTTTGGTTT	62	169
TLR22-3-R	CTGTGTTCAGGAATGCCTTG		
IRF3-F	CCAAACCACACCATCCAATCT	62	109
IRF3-R	ACTACCTGTTCCTGACGGTATC		
MyD88-F	GAGTTTCCCACTCCGTTAAGA	62	92
MyD88-R	CGCCGAGATGATGGACTTTA		
β-actin-F	GACCACCTTCAACTCCATCAT	62	126
β-actin -R	GTGATCTCCTTCTGCATCCTATC		

## Data Availability

Publicly available datasets were analyzed in this study. The rest of the data presented in this study are available on request from the corresponding author.

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
