# Peer review of "Molecular Cloning of Toll-like Receptor 2 and 4 (SpTLR2, 4) and Expression of TLR-Related Genes from Schizothorax prenanti after Poly (I:C) Stimulation"

_genes, 2023, doi:10.3390/genes14071388_

Round 1
Reviewer 1 Report
The present manuscript describes cloning of two toll-like receptor genes from a fish species Schizothorax prenanti.
In general the results have some merit, but they definitely require a more sound presentation.
=== Major text presentation issues
In general, the authors should clearly say what this manuscript adds to the previous ones on the topic (for example, https://doi.org/10.3390/genes13101862 or https://doi.org/10.1016/j.fsi.2019.10.027). Basically, what am I asking is: how clear is the overall picture with this gene family now? How many TLR genes remain unexplored?
It is very confusing that the previous works on S. prenanti TRLs are not explained in the introduction.
In addition, the title of the paper only describes half of its results, making it even more difficult to navigate between the multiple papers on TLRs of S. prenanti. It it very general and sounds more like a review. Please consider making it more precise.
Then, the authors mention transcriptome sequencing but do not reference it, at least not explicitly. Is it published? Why do the authors undertake the cloning procedure if they have the transcriptome assemblies? Are the transcripts incomplete? This should be explained in more details.
I totally fail to see the value of Figures 1 and 3. I cannot imagine a researcher using these pictures to get the sequences. This being said, I’d like to commend the authors on submitting the sequences to Genbank. Unfortunately, it looks like they are not processed yet. It would be even better to add a fasta or genbank file as supplementary information (instead of figures).
The last part of the result requires summing up (there or in discussion): which genes are inducible and which are not? How does it correlate to LPS exposure from the previous manuscript?
More on this: L27 says: “Our findings indicate that TLR4 is expressed in S. prenanti and”. Why do the authors mention exactly TLR4 here? Is TLR2 not expressed?
=== Important methodology or presentation issues
The quality of all figures (except 3) is very low. They are barely readable. Please take care of this issue!
L237-240: “We analyzed the phylogeny of the SpTLR2 and SpTLR4 amino acid sequences to determine the evolutionary relationships between S. prenanti and other vertebrates based on sequences in the GenBank
database (Fig. 6).”: please explain the idea of this analysis. I do not agree that TLR2 and TLR4 phylogenies can be used to explore and reveal the relationships between species, nor do I think it is necessary to explore the phylogeny of the species. Right now I can only see that SpTLR2 is a representative of TLR2, and SpTLR4 is a representative of TLR4. If the purpose was to confirm the subfamily for the newly sequenced genes, please state it clearly. If the purpose was to compare the evolutionary history of genes and species, other analyses are needed, like adding species phylogeny based on a marker gene (for example, cytochrome B).
The Methods section states that the qPCR data were processed in the same way, but Figs 8 and 9 are different (either relative to actin or relative to the control sample). Please state more clearly how you analyzed the data.
Speaking of qPCR further, was false discovery rate controlled for the number of tests?
It is possible to add the raw qPCR data as a supplement for reproducibility? Was amplification efficiency measured?
Was Ct controlled to be in the measurable range?
Furthermore, maybe the author could consider at some point aggregating their qPCR data for this species and analyzing it together / adding it to the GEO database. Maybe it would be great for a review of the TLR family when all of the necessary genes are explored.
L163: “Positive bacterial clones were sequenced”: was sequencing performed from the same primers or from standard primers annealing to the vector?
Were only two primers used? It seems not very probable to get high-quality 1200 bp-long sequences...
=== Minor text issues
L14 “The SpTLR2 and SpTLR4 was cloned and identified” => “The SpTLR2 and SpTLR4 were cloned and identified”
L21: it’s not homology, it’s identity
L57-58: “pathogenic microorganisms, including bacteria, viruses, and parasites”: eukaryotic parasites? Please clarify
L98-99: “to bacterial infections, such as Aeromonas hydrophila [32,33], Streptococcus agalactiae [34], and reoviruses [35],”: change to “to bacterial infections, such as Aeromonas hydrophila [32,33] or Streptococcus agalactiae [34], as well as reoviruses [35],”
L122: “60 × 30 × 40 cm3”: “60 × 30 × 40 cm”?
How many fish per tank?
Figure 2: maybe add info which of these are fish species?
Figure 6: it would be great to show (for example, with colored background) which part of the tree is TLR2 and TLR4.
Figure 7: “under normal conditions”
L440-441: “Our results support the conclusion that MyD88 plays a crucial role in the innate immunity of fish”
L524-525: “our findings demonstrate that S. prenanti expresses functional TLR2 and TLR4 proteins”: only indirectly, you did not measure the proteins or their function in any way.
The language is generally fine and understandable, it only requires minor check before the manuscript is finalized.
Reviewer 2 Report
The manuscript titled “Molecular cloning and expression analysis of Toll like receptor-related genes from Schizothorax prenanti” deals with the molecular characterization and gene expression of SpTLR2 and SpTLR4 in one species of economic interest that is susceptible to pathogens in aquaculture. The research provides interesting insights into the immune defence system in Schizothorax prenanti, especially after induction of antiviral response by poly (I:C) challenge.
Overall, the study is well-planned, and the organization of the different sections is consistent; however, there are still margins for improvement.
The Introduction report on the aims of the study: why is there a great emphasis only for TLR4 when the molecular characterization includes even TLR2? This should be clarified.
All the 8 figures are of bad quality: please verify the resolution required by the journal guidelines and make them easy to read. Figures 1 and 3 are not necessary, except that the authors can integrate the information on secondary structure and domains in the nucleotide/amino acid sequences.
I suggest rephrasing some parts of the Results section; you should not explain what a figure depicts “Figure 7 shows the ubiquitous but variable expression of SpTLR2 and SpTLR4 transcripts in all eight tissues” (lines 297-298) but instead it should support the obtained data (as it is for figures 1-6). Please, rephrase 3.2 section and the subsequent paragraphs, accordingly.
Minor points
Line 14: “was cloned” should be “were”
Line 127: “twice daily” should be twice a day
Line 162: “Escherichia coli” should be in italics.
Line 336: delete the full stop in “at 24 h) . but no “
English language is of good quality.
Round 2
Reviewer 1 Report
I’d like to thank the authors for taking the questions and suggestions seriously and clarifying their text.
I have one suggestion remaining, which is about point 12 (“Positive bacterial clones were sequenced”...) in the first review.
The author did clarify the issue a little bit but I’m still puzzled. As the sequence length was > 800 bp, walking primers should have been designed, but their sequences were not provided.
And even more importantly, it would be great to add this information into the manuscript as well.
There are still minor language-related issues remaining, so the text could benefit from one more round of proofreading.
Some examples below (I definitely couldn’t find them all with a quick look):
L. 28: “Our findings indicate that TLR2 and TLR4 is” => “Our findings indicate that TLR2 and TLR4 are”
L. 128-130: “The fish were maintained in glass tanks with a volume of (60 × 30 × 40) cm3, 12 fish are placed per tank, and with aerated tap water at a temperature of 20 ± 1°C.” => for example, “The fish were maintained in glass tanks with a volume of (60 × 30 × 40) cm3 with aerated tap water at a temperature of 20 ± 1°C; 12 fish were placed per tank”